# IFN-Inducible SerpinA5 Triggers Antiviral Immunity by Regulating STAT1 Phosphorylation and Nuclear Translocation

**DOI:** 10.3390/ijms24065458

**Published:** 2023-03-13

**Authors:** Congcong Wang, Yajie Liu, Xinglai Liu, Jin Zhao, Bing Lang, Fan Wu, Ziyu Wen, Caijun Sun

**Affiliations:** 1School of Public Health (Shenzhen), Sun Yat-sen University, Shenzhen 518107, China; 2Key Laboratory of Tropical Disease Control (Sun Yat-sen University), Ministry of Education, Guangzhou 510080, China

**Keywords:** SerpinA5, innate immunity, interferon-stimulated genes (ISGs), STAT1 phosphorylation

## Abstract

Deeply understanding virus-host interactions is a prerequisite for developing effective strategies to control frequently emerging infectious diseases, which have become a serious challenge for global public health. The type I interferon (IFN)-mediated JAK/STAT pathway is well known for playing an essential role in host antiviral immunity, but the exact regulatory mechanisms of various IFN-stimulated genes (ISGs) are not yet fully understood. We herein reported that SerpinA5, as a novel ISG, played a previously unrecognized role in antiviral activity. Mechanistically, SerpinA5 can upregulate the phosphorylation of STAT1 and promote its nuclear translocation, thus effectively activating the transcription of IFN-related signaling pathways to impair viral infections. Our data provide insights into SerpinA5-mediated innate immune signaling during virus-host interactions.

## 1. Introduction

The innate immune system serves as the first line of host defense against viral infections by sensing the structurally conserved pathogen-associated molecular patterns (PAMPs) [1,2]. Following the specific recognition between various PAMPs and cytoplasmic/endosomal pattern recognition receptors (PRRs), interferons (IFNs) are frequently induced in response to microbial infections [3,4]. IFNs can bind to and dimerize the IFN receptors, and then the IFN receptor-associated Janus kinase-transcriptional signal transducer and activator (JAK-STAT) pathway is activated to induce a variety of IFN-stimulated genes (ISG) to further modulate the antiviral immunity [5,6]. Recently, we have reported that some ISGs, including KMO, TAP1, and CH25H, played a direct antiviral function by different mechanisms [7,8,9]. Nevertheless, novel ISGs with potential antiviral roles should be further identified, which will be helpful in developing effective therapeutics against the frequently emerging and re-emerging infectious diseases [10].

Human serpin peptidase inhibitor clade A member 5 (SerpinA5), also named as protein C inhibitor (PCI), alpha-1 antiproteinase, antitrypsin, was first discovered as an inhibitor of the anticoagulant protease activated protein C (APC) [11]. As a secreted protein with an extensive tissue distribution, SerpinA5 has been found in a variety of body fluids, including blood plasma, seminal plasma, and cervicovaginal fluid [12]. SerpinA5 belongs to a family member of glycoproteins, which can suppress several serine proteases, plasminogen activators, and kallikreins [13]. Therefore, it is traditionally thought to play roles in thrombosis and hemostasis in multiple organs. In addition, SerpinA5 might act as an antimicrobe agent and a tumor suppressor to negatively regulate tumor cell growth and metastasis [14,15,16]. Recently, a higher level of SerpinA5 was observed in the cervicovaginal fluid in HIV-1-exposed seronegative individuals than that in HIV-1-exposed seropositive individuals [17], suggesting that SerpinA5 might be a protective factor when determining susceptibility to HIV-1 infection. So far, to our best knowledge, the roles of SerpinA5 in regulating innate immune signaling to control viral infection are not clarified.

In this study, we thus explored the novel antiviral function of SerpinA5, which is independent of its previously reported functions, and further studied the involved mechanism by regulating innate signaling pathways. This comprehensive study will contribute to the clarification of the biological role of SerpinA5, by deeper understanding the subtle virus-host interactions, and thereby provide insights to develop novel antiviral therapeutics.

## 2. Results

### 2.1. SerpinA5 Is an IFN-Stimulated Gene with Antiviral Function

To determine whether SerpinA5 is an IFN-stimulated gene, we detected the expression level of SerpinA5 in different monocytes/macrophage cell lines, including mice bone marrow-derived macrophages (BMM), Raw 264.7 cells, and THP-1 cells, in response to multiple kinds of Toll-like receptor (TLR) agonists (LPS, PolyI:C, R848), interferon α (IFN α), and HSV-1 infections [8]. Our results showed that SerpinA5 expression in BMM cells was significantly upregulated after the stimulation with TLR agonists and viral infections in a time-dependent manner (Figure 1A–E). Similar results were also observed in Raw 264.7 cells and THP-1 cells (Appendix A). This elevated expression of SerpinA5 protein in response to TLR agonists and HSV-1 infections was further confirmed in BMMs by Western blotting assay. However, this upregulation was abrogated in IFN α receptor-deficient BMM cells ((IFNAR−/−) BMM) (Figure 1F), indicating that SerpinA5 expression was IFN-dependent.

Next, we investigated whether SerpinA5 could exert an unrecognized role in antiviral innate immunity. Hela cells and A549 cells were transfected with increasing amounts of SerpinA5-expressing plasmids, and then infected with HSV-1. Results demonstrated that SerpinA5 overexpression in Hela cells significantly suppressed HSV-1 replication by quantifying the expression of SerpinA5 and HSV-1 UL27 by Western blotting (Figure 1G) and RT-qPCR (Figure 1H). The direct inhibition of viral replication by SerpinA5 was further confirmed by the 50% tissue culture infectious dose (TCID50), and the virus titers in SerpinA5-treated samples were significantly decreased in a dose-dependent manner (Figure 1I). Similar results were also observed in SerpinA5-treated A549 cells (Figure 1J–L).

To further verify the inhibitory effect of SerpinA5 on viral replication, we designed and synthesized the corresponding small interfering RNA (siRNA) to downregulate the expression of SerpinA5. We constructed a 293T cell line with stably expressing SerpinA5 (293T-SerpinA5 cell) for this study (Appendix A). The knockdown efficiency of the siRNAs was evaluated by RT-qPCR and Western blotting analysis (Figure 1M,N), showing that the siRNA-820 had a high knockdown efficiency. Then, SerpinA5 siRNA-820 was used for the subsequent assays. As expected, these SerpinA5-knockdown-cells became more susceptible to HSV-1 infection by RT-qPCR and Western blotting analysis, when compared to the mock-treated 293T-SerpinA5 cells (Figure 1O,P). Together, these results indicated that SerpinA5 had an important antiviral function.

### 2.2. SerpinA5 Can Play the Antiviral Function through Modulating IFN Signaling Pathways

To clarify the underlying mechanism for the antiviral activity of SerpinA5, we subsequently performed an unbiased transcriptomic analysis of the expression landscape and cellular responses in response to SerpinA5 treatment. A549 cells were transfected with SerpinA5-expressing plasmid or mock vector for 24 h. Then, these cells were infected with HSV-1 for 12 h and the total RNA was collected. Transcriptome analysis was then carried out by RNA sequencing. Our analysis revealed a marked alteration in gene expression with 1099 upregulated genes and 894 downregulated genes (Figure 2A). KEGG analyses also revealed the enrichment of differentially expressed genes (DEGs) in the TNF signaling pathway, viral protein interaction with cytokine and cytokine receptor, and Toll-like receptor signaling pathway (Figure 2B). The analysis of gene ontology (GO) annotation further demonstrated that SerpinA5-mediated DEGs were mainly involved in type I IFN, TNF, viral protein interaction with cytokine and cytokine receptor, and Toll-like receptor signaling (Figure 2C–F), suggesting a potential role of SerpinA5 in the regulation of IFN signaling and viral responses. Importantly, this observation was further confirmed by the promoter luciferase reporter system, and we found that SerpinA5 overexpression significantly promoted the activation of IFN-β promoter and IFN-stimulated response element (ISRE) promoter (Figure 2G,H). Furthermore, we further confirmed the upregulated expression of IFN-related genes after SerpinA5 treatment by RT-qPCR, including IFN-β, MX1, CXCL10, IL-1β, and IFN-λ (Figure 2H–L). In addition, consistent with the above result, the expression of SerpinA5 in these experiments significantly impaired HSV-1 replication (Figure 2M). Taken together, these results suggested that SerpinA5 played an antiviral function through modulating IFN-related signaling pathways.

### 2.3. SerpinA5 Activated IFN Production Independent of cGAS-STING Signaling Pathway 

Next, we further explored the underlying mechanism through which SerpinA5 activated IFN production to play a role in antiviral activity. Since SerpinA5 promoted the DNA virus (HSV-1)-triggered activation of IFN-β promoter and ISRE promoter, we first detected whether SerpinA5 interacted with the cGAS-STING signaling pathway [18]. A series of plasmids expressing cGAS, STING, TBK1, IRF3, or IRF7, along with ISRE luciferase reporter plasmid and the internal control plasmid pRL-TK, were cotransfected with or without SerpinA5-expressing plasmid, and then the activation of luciferase activity was evaluated. As expected, these components of the cGAS-STING signaling pathway effectively activated the ISRE reporter system (Figure 3). Intriguingly, SerpinA5 overexpression further promoted the activation of the ISRE reporter system, on the basis of the stimulation of cGAS, STING, TBK1, IRF3, or IRF7 (Figure 3A–E), indicating that SerpinA5 might activate IFN production by targeting the downstream of IRF3 and IRF7. To verify our hypothesis, we conducted the coimmunoprecipitation assay to further test the potential interaction between SerpinA5 and the key molecules of the cGAS-STING signaling pathway. The results showed that SerpinA5 did not interact with cGAS, STING, TBK1, IRF3, or IRF7 (Figure 3F). Moreover, we blocked the cGAS-STING signaling using C-176, which is a kind of STING inhibitor. Results showed that SerpinA5 overexpression could upregulate the expression of IFN-β and ISG56 even in presence of C-176 treatment (Figure 3G). These findings imply that SerpinA5 can activate the IFN production independent of cGAS-STING signaling pathway.

### 2.4. SerpinA5 Induced Antiviral Innate Immunity by Promoting STAT1 Phosphorylation and Nuclear Translocation 

Except for the cGAS-STING pathway, we further explored how SerpinA5 activated the IFN production in other molecular mechanisms, such as the JAK-STAT pathway. Considering that STAT1 nuclear translocation is an essential step to activate the downstream genes of IFN signaling cascades, we investigated whether SerpinA5 regulates JAK-STAT cascades. A549 cells were transfected with or without SerpinA5-expressing plasmid, followed by IFN-β stimulation, and then the levels of STAT1 and phosphorylated-STAT1 were quantified by Western blotting assay. Results demonstrated that SerpinA5 overexpression significantly promoted the phosphorylation of STAT1 (Figure 4A). In contrast, SerpinA5 knockdown decreased the phosphorylation of STAT1 (Figure 4B). Using confocal microscopy, we also detected the subcellular localization of STAT1 protein after IFN-stimulation in the vector or overexpression of SerpinA5 protein. Consistent with previous studies, STAT1 was mainly distributed in the cytoplasm before IFN-β stimulation, and then transported into the nucleus after IFN-β stimulation. Interestingly, SerpinA5 overexpression significantly promoted STAT1 nuclear translocation (Figure 4C). We also verified that SerpinA5 could interact with STAT1 but could not interact with STAT2 and IRF9 (Figure 4E–G). In addition, changes in STAT1 phosphorylation and nuclear translocation in response to SerpinA5 stimulation were also confirmed in the nuclear and cytoplasmic fractions, respectively, at the protein level by Western blotting analysis (Figure 4H). Overall, these results demonstrated that SerpinA5 might induce antiviral innate immunity by promoting STAT1 phosphorylation and nuclear translocation.

## 3. Discussion

The frequently emerging and re-emerging outbreaks due to highly pathogenic viruses are a severe crisis for global public health and social security. Deeply understanding virus-host interactions is a prerequisite for developing effective strategies to control viral infections. As the key signal pathway against viral infections, the Type I IFN-mediated JAK/STAT pathway plays an essential role in host antiviral immunity [19,20,21,22]. The JAK/STAT pathway has been well-studied, but the exact regulatory mechanisms by various ISGs are not yet fully understood. In this study, we reported for the first time that SerpinA5 can play a novel antiviral function as an IFN-stimulated gene, and we also provided evidence that SerpinA5 can upregulate the phosphorylation of STAT1, and thus promote the formation of a STAT1/STAT2 complex and its nuclear translocation to effectively activate the transcription of IFN-related signaling pathways to impair viral infections.

SerpinA5 is described initially as an inhibitor of anticoagulant protease-activated protein C (APC) and plasminogen activator urokinase [23], and its inhibitory activity can be regulated by binding to glycosaminoglycans, phospholipids, and retinoic acid [24]. Previous studies also revealed that SerpinA5 might play a function in host defense, such as tumor suppression. Interestingly, a recent study showed that SerpinA5 might have a protective role against HIV-1 infection [17]. In addition, Serpin C1 (antithrombin III) was also reported to exhibit antimicrobial activity [25]. Of note, the latest study demonstrated that the rs2093266 and rs1955656 polymorphisms in SerpinA4 and SerpinA5 genes might be risk factors for COVID-19-induced AKI (acute kidney injury) [26]. Further studies revealed that the antimicrobial activity of SerpinA5 might be attributed to its heparin-binding sequence (H-helix), and the nuclear localization signal in the H-helix was responsible for the nuclear translocation of SerpinA5 [27,28], implying that SerpinA5 might have a function in transcriptional regulation. However, it is not clarified yet how SerpinA5 regulates the innate immune signaling against viral infections.

We therefore investigated the relationship between SerpinA5, HSV, and TLR agonists to confirm SerpinA5 as an IFN-stimulated gene. Based on these observations, we subsequently explored whether SerpinA5 could affect the outcome of virus infections in cell culture. The results from overexpression and knockdown assays showed that SerpinA5 could remarkably inhibit the replication of HSV-1. Further, luci-reporter results also implied that the viral inhibition of SerpinA5 may be associated with IFN signaling pathways, revealing a novel link between SerpinA5 and the innate immunity. Since our results suggested that SerpinA5 might play an antiviral function through modulating IFN-related signaling pathways. Next, we performed RNA-seq assay to further identify the molecular mechanism through which SerpinA5 activated IFN production, and we explored the possible signaling pathways, including cGAS-STING, JAK-STAT, RIG-I, TLR, and other pathways. Interestingly, our results showed that SerpinA5 did not interact with the cGAS-STING pathway. Alternatively, we found that SerpinA5 activated the IFN production through the JAK-STAT pathway. The importance of STAT1/2 in viral infection has been well-illustrated. After IFN-α/-β binding with IFNAR, STAT1/2 can be phosphorylated and translocated to the nucleus, where they assemble with IRF9 to form IFN-stimulated gene factor 3 (ISGF3). ISGF3 then binds to IFN-stimulated response elements (ISREs). Consistent with our results, several studies demonstrated that SerpinA5 translocated to the nucleus and exhibited a positive effect on the nuclear translocation [29]. Therefore, further investigation will focus on elucidating how STAT1 and SerpinA5 are involved in binding to other transcription factors.

In the present study, we found that Serpin5 was a positive regulator in activating the IFN-related signaling pathways. For example, SerpinA5 treatment can upregulate the expression of IFN-related genes including IFN-β, MX1, CXCL10, IL-1β, and IFN-λ. Mechanistically, similar to other ISGS (RIG-1, MDA-5, STATs, TAP1, etc.) [30,31], SerpinA5 can regulate the PRRs, IFN-a/-β, and JAK/STAT together to form a positive feedback loop to promote IFN production. Consequently, the secreted IFNs bind to type I IFN receptors (IFNAR1 and IFNAR2) and activate JAK family kinases to regulate the expression of more ISGs. It is well known that JAK1 and TYK2 are activated by receptor engagement to phosphorylate the intracellular domains of the IFNα/β receptors, and thus provide the recruitment sites for the STAT1 and STAT2 SH2 domains [19,23,32]. Besides, SerpinA5 can translocate to the nucleus to promote ISGF3 complex nucleus translocation [29,33]. The next study should further investigate which domain of SerpinA5 protein will contribute to its antiviral function, and how SerpinA5 protein can affect the form of STAT1 homodimer or STAT1/STAT2 heterodimer complex in molecular mechanisms.

In summary, we identified that SerpinA5 could exert a previously unrecognized function as a novel ISG with antiviral activity, and we also proposed a working model for the roles of SerpinA5 in regulating innate immune signaling to control viral infection (Figure 5). Our data provide insights into the deeper understanding of SerpinA5-mediated innate immune signaling during virus-host interactions.

## 4. Materials and Methods

### 4.1. Cells and Virus

HEK-293T cells, A549 cells, Hela cells, and RAW 264.7 cells were cultured in complete Dulbecco’s modified Eagle’s medium (DMEM, Gibco, Grand Island, NY, USA) containing 10% fetal bovine serum (FBS, Gibco) and 1% penicillin/streptomycin (Gibco). THP-1 cells were cultured in a conditioned RPMI 1640 medium containing 10% fetal bovine serum (FBS, Gibco), and 1% penicillin/streptomycin (Gibco). Wild-type bone-marrow-derived macrophage cells (WT-J2-BMM) and interferon-α receptor-deficient cells (Ifnar−/−-J2-BMM), gifted by Prof. Genhong Cheng (UCLA, Los Angeles, CA, USA), were cultured in conditioned RPMI 1640 medium containing 10% fetal bovine serum (FBS, Gibco, Grand Island, NY, USA), 1% penicillin/streptomycin (Gibco, Grand Island, NY, USA), 10 mM HEPES (pH 7.8), and 1% M-CSF (Gibco, Grand Island, NY, USA). These cells were all cultured at 37 °C in an atmosphere of 5% of CO_2_. In order to obtain the cell line stably expressing SerpinA5, the recombinant lentivirus carrying the Human SerpinA5 gene was constructed and used to infect HEK-293T cells. The cell line stably expressing this SerpinA5 was selected by adding Puromycin and validated by RT-PCR and Western Blotting analysis. HSV-1 strain F carrying green fluorescent protein (GFP)- Renilla luciferase (Luc) reporter was stored in our laboratory. 

### 4.2. Plasmid Constructs

Human SerpinA5 cDNA was amplified from A549 cells and subcloned into a pcDNA3.1 vector with N-terminal tagged Flag (pcDNA3.1-FLAG-SerpinA5). The primers used for PCR were as follows: 5’-CCGGAATTCATGCAGCTCTTCCTCCTC-3’ (sense) and 5’-CGCGGATCCTCAGGGGCGGTTCACTTT-3’ (antisense). The plasmids expressing the components of the type I IFN pathway, including cGAS, STING, TBK1, IRF3, and IRF7 were stored in our lab. 

### 4.3. Transient Transfection and siRNA Experiment

According to the manufacturer’s manual, the cells with approximately 80% confluence were transfected with the corresponding plasmids with Lipofectamine 2000 transfection reagent. After 5 h of transfection, replace the medium with DMEM or 1640 containing 5% FBS and 1% penicillin/streptomycin, and then continue to culture the cells for 24 h. Three siRNAs targeting SerpinA5 and negative control siRNA were synthesized by Sangon (Shanghai, China). According to the manufacturer’s agreement, 100 nM of designated siRNAs were transfected into cells for 48–72 h with LipofectamineTM RNAiMax transfection reagent (Invitrogen, Carlsbad, CA, USA). The knockdown effect of the targeted gene was detected by quantitative real-time PCR (RT qPCR) or Western blotting analysis. The sequences of all siRNAs were listed in Appendix A.

### 4.4. Western Blotting Analysis

The Western blotting assay was performed as we previously described [7]. Antibodies against the FLAG tag (M10001), P-STAT1-Y701(T66917), and Antibodies against STAT1 (T55227) were purchased from Abmart (Shanghai, China). Antibodies against SerpinA5(ab154275) and GAPDH (ab9485) were purchased from Abcam (Cambridge, UK). Antibody against UL27(H1142) was purchased from Santa Cruz Biotechnology (Santa Cruz, CA, USA).

### 4.5. Quantitative Real-Time PCR

The total RNA was extracted by using an RNA extraction kit (EZBioscience, Shanghai, China) according to the manufacturer’s manual. The cDNA was synthesized with Hifair III 1st strand cDNA (Yeasen, Shanghai, China) and mixed with PerfectStart SYBR Green SuperMix (TransGen Biotech, Beijing, China), followed by quantification of gene expression with the BioRad CFX Connect system. The primer sequences used in our study are described in Appendix A.

### 4.6. RNA-Seq Library Preparation, Sequencing, and Data Processing

RNA-seq experiments were performed as previously described [34,35]. In brief, A549 cells were transfected with or without SerpinA5-expressing plasmid at a dose of 2 μg for 24 h. Then, the cells were infected with HSV-1 (MOI = 0.5) for another 12 h. These samples were collected for the total RNA extraction and then used to generate RNA-seq libraries with a TruSeq PE Cluster Kit v4-cBot-HS (Illumina, San Diego, CA, USA) according to the manufacturer’s instructions. The raw data obtained by Illumina HiSeq sequencing was filtered, and the clean reads were aligned to the reference sequence. According to their expression levels in different sample groups, these genes with *p*-values < 0.05 and fold changes > 2 (i.e., log2FC > 1) were considered differentially expressed genes (DEGs). GO function analysis, pathway function analysis, and cluster analysis were performed. The volcano plot was drawn by Volcano mapping tool in SangerBox, and the heatmap was drawn by the R-Studio.

### 4.7. TCID50

The cells were transfected with 0, 0.5, 1, or 2 μg of SerpinA5-expressing plasmids for 24 h and then infected with HSV-1 (MOI = 0.5) for 12 h. The supernatant was collected, and the total virus titer was determined by the 50% tissue culture infectious dose (TCID50). Briefly, Vero cells were seeded into 96-well plates at a density of 104 cells/well. After 24 h, the above supernatant was serially diluted 10-fold to 10–8 using DMEM containing 2% FBS. Then, 100 μL per well of the dilution was added to a 96-well plate. The 96-well plate was incubated at 37 °C and 5% CO_2_ atmosphere, and the viral cytopathic effect was observed every 24 h. The TCID50 was calculated using Reed and Muench formula.

### 4.8. Co-Iimmunoprecipitation

The cells were harvested using NP40 lysis buffer containing protease inhibitors (Beyotime, Shanghai, China). After incubation on ice for 30 min, lysates were centrifuged at 13,000 rpm for 20 min at 4 °C. Subsequently, a part of the cell lysate was analyzed as input, and the other part of the cell lysate was precipitated overnight with IgG and Flag antibody, respectively, at 4 ° C. The next day, the beads were washed three times with wash buffer (50 mM Tris pH 7.4, 500 mM NaCl, 0.1% (*v*/*v*) NP-40, 1 mM EDTA). The precipitated proteins were eluted from beads or gel by heating the sample in SDS loading buffer at 100 °C for 10 min. The precipitates were subjected to Western blot.

### 4.9. Luciferase Assay

The Steady-Glo^®^ Renilla Luciferase detection system and Dual-Luciferase Reporter Assay were employed according to the manufacturer’s instructions (Promega, Madison, WI, USA).

### 4.10. Confocal Microscopy

Cells were seeded into NuncTM glass bottom dishes and transfected with various plasmids using LipofectamineTM2000 according to the manufacturer’s instructions. After 24 h transfection, the cells were stimulated with IFN-β for 30 min and then fixed with 4% paraformaldehyde. After blocking with 5% bovine serum, the cells were incubated with appropriate primary antibodies against the FLAG tag and STAT1 and fluorochrome-conjugated secondary antibodies. Nuclei were stained with 4,6-diamidino-2-phenylindole dihydrochloride (DAPI) (Solarbal). The cells were visualized using confocal microscopy (Nikon Microsystems, Tokyo, Japan). All images were captured and processed by Nikon element viewer.

### 4.11. Quantification and Statistical Analysis 

Statistical analyses were performed using GraphPad Prism software version 8 (GraphPad Software, La Jolla, CA, USA). Statistical significance was calculated using Student’s two-tailed unpaired *t*-test or ANOVA with Holm–Sidak’s multiple comparisons test. * *p* < 0.05; ** *p* < 0.01; *** *p* < 0.001.

## Figures and Tables

**Figure 1 ijms-24-05458-f001:**
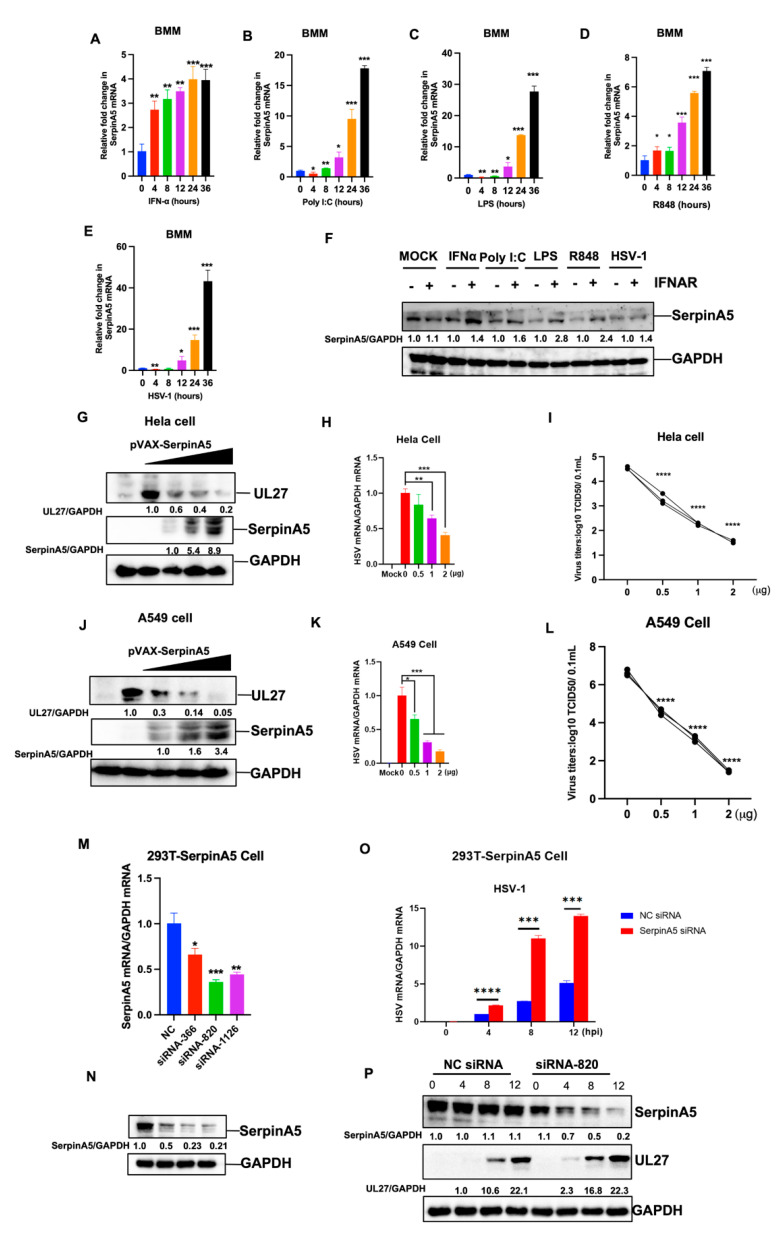
SerpinA5 is an IFN-stimulated gene with antiviral function. (**A**–**E**) BMM cells were stimulated with IFN-α (2000 U/mL), Poly I:C (25 μg/mL), R848 (100 nM), and HSV-1 (MOI = 0.5), respectively, and then the expression of SerpinA5 was detected by qPCR. The expression level of mRNA was normalized to the expression of GAPDH, and the data from at least triplicates were shown as the mean ± SD. * *p* < 0.05, ** *p* < 0.01, *** *p* < 0.001. (**F**) Wild-type bone marrow-derived macrophage (WT-J2-BMM) cells or interferon α receptor-deficient (IFNAR−/−)-J2-BMM cells were stimulated with IFN-α (2000 U/mL), poly I:C(25 μg/mL), R848 (100 nM), and HSV-1 (MOI = 0.5) for 24 h, respectively, and then the expression of SerpinA5 protein was measured by Western blot analysis. GAPDH was used as an internal control. The relative ratios of SerpinA5 and GAPDH were marked at the bottom of the pictures. (**G**–**I**) Hela cells were transfected with 0, 0.5, 1 or 2 μg of Pvax-SerpinA5-expressing plasmids for 24 h. The cells were then mock-infected or infected with HSV-1 (MOI = 0.5) for 12 h. The expression of viral proteins (**J**) and viral RNA (**K**) were detected by Western blotting and qPCR, respectively. The virus yields were measured by TCID50 assay (**L**). (**J**) A549cells were transfected with 0, 0.5, 1 or 2 ug of Flag-SerpinA5-expressing plasmids for 24 h. The cells were then mock-infected or infected with HSV-1 for 12 h. The expression of viral proteins and viral RNA were detected by Western blotting and qPCR, respectively. (**M**,**N**) 293T-SerpinA5 cells were transfected with 120 nM of negative control (NC) siRNA or SerpinA5 siRNA (siRNA-366, siRNA-820 siRNA-1126) for 24 h, the knockdown efficiency of each siRNA was then evaluated by qPCR and Western blotting analysis, respectively. (**O**) 293T-SerpinA5 cells were transfected with 120 nM of NC siRNA or SerpinA5 siRNA (siRNA-820) for 24 h, and the cells were then infected with HSV-1 (MOI = 0.5) for 0, 4, 8 or 12 h. The expression of viral proteins and viral RNA (**P**) were detected by Western blotting and qPCR, respectively. Data from at least triplicates were shown as the mean ± SD. * *p* < 0.05, ** *p* < 0.01, *** *p* < 0.001, *****, p* < 0.0001.

**Figure 2 ijms-24-05458-f002:**
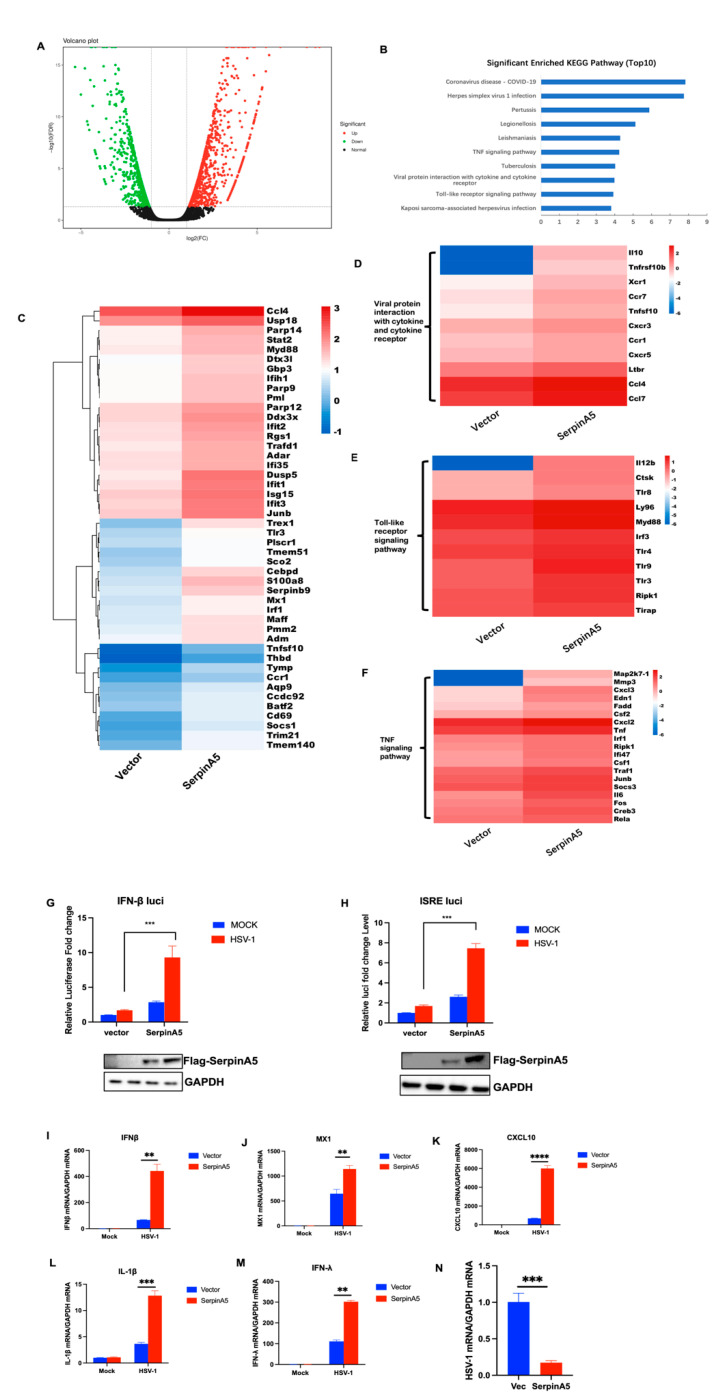
SerpinA5 played an antiviral function through modulating IFN-related signaling pathways. A549 cells were transfected with 2 μg SerpinA5-expressing plasmid or mock vector for 24 h and then infected with HSV−1. Transcriptome analysis was then carried out by RNA sequencing. (**A**) The representative differentially expressed genes (DEGs) related with innate immunity in response to SerpinA5 treatment during HSV-1 infection, based on two independent experiments. (**B**) Top10 significantly enriched signaling pathways identified with KEGG Orthology-Based Annotation System (KOBAS) 2.0. (**C**–**F**) Heatmap of DEGs contributing to the GO terms in response to type I IFN, TNF, Viral protein interaction with cytokine and cytokine receptor, and Toll-like receptor signaling pathway. Color code represents the log2 fold change compared to NHT. (**G**,**H**) A549 cells were transfected with 150 ng of SerpinA5–expressing plasmids together with 100 ng of IFN-β promoter (**G**), ISRE promoter (**H**) luciferase reporter plasmids and 10 ng of the internal control plasmid pRL-TK for 24 h, and the cells were mock-infected or infected with HSV-1 for another 6 h. The luciferase activity was determined by the dual-luciferase assay. (**I**–**M**) A549 cells were transfected with 0 or 1 μg of Flag-SerpinA5-expressing plasmids for 24 h, and then mock-infected or infected with HSV-1 for 12 h. The expression level of IFN-β, MX1, CXCL10, IL-1β, and IFN-λ was analyzed by RT-qPCR. The expression of HSV-1 viral RNA in SerpinA5-transfected A549 cells was also detected by RT-qPCR (**N**). Data (**G**–**N**) from at least triplicate experiments were shown as the mean ± SD. ** *p* < 0.01, *** *p* < 0.001, **** *p* < 0.0001.

**Figure 3 ijms-24-05458-f003:**
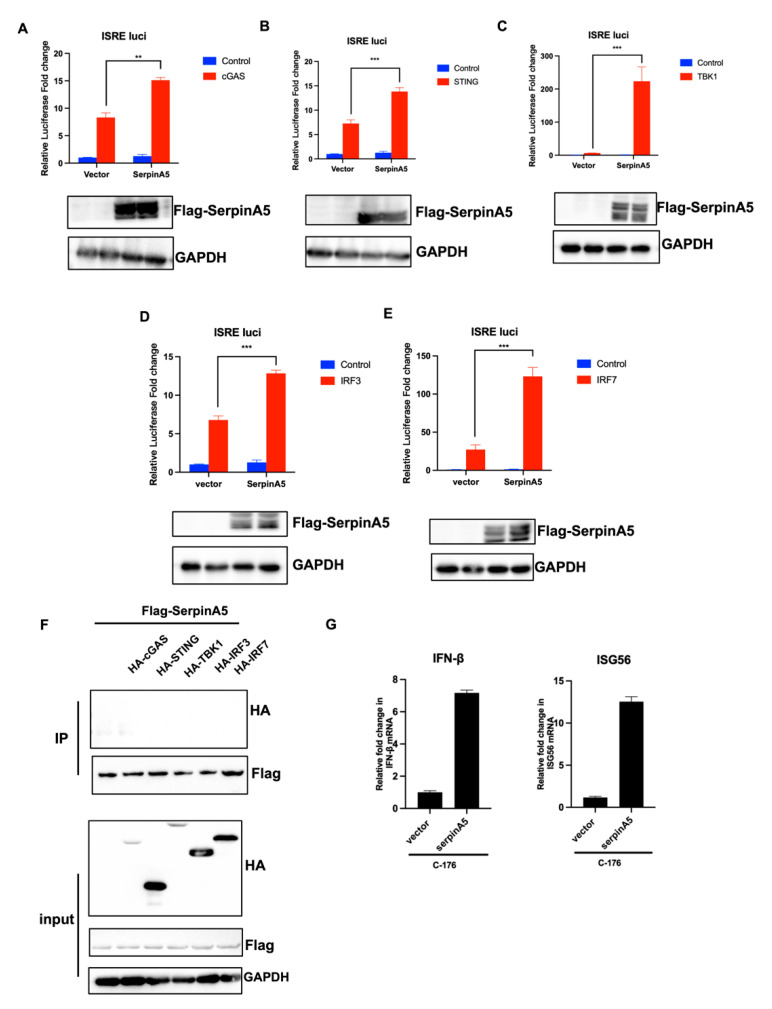
SerpinA5 activated IFN production independent of cGAS-STING signaling pathway. (**A**–**E**) A549 cells were cotransfected with 8 μg empty vector or 4 μg FLAG- SerpinA5-expressing plasmids and HA-tagged cGAS-, STING-, TBK1-, IRF3-, or IRF7-expressing plasmids with ISRE luciferase reporter plasmid and the internal control plasmid pRL-TK. The luciferase activity was determined at 24 h post transfection. Data from at least triplicate experiments were shown as the mean ± SD. ** *p* < 0.01, *** *p* < 0.001. (**F**) HEK-293T cells were transfected with 8 μg vector or FLAG-SerpinA5-expressing plasmids and 4 μg HA- tagged cGAS-, STING-, TBK1-, IRF3-, or IRF7-expressing plasmids for 36 h. The cells were collected and subjected to immunoprecipitation experiments and immunoblotting analysis. The lysates were immunoprecipitated using anti–HA antibody and subjected to Western blotting analysis using the indicated antibodies. (**G**) Raw264.7 were transfected with vector and serpinA5-expressing plasmids for 12 h, treated with C-176 0.2μM for 4 h, and then infected with HSV-1 for 6 h. The cells were collected and the expression levels of IFNβ and ISG56 RNA were determined by qPCR.

**Figure 4 ijms-24-05458-f004:**
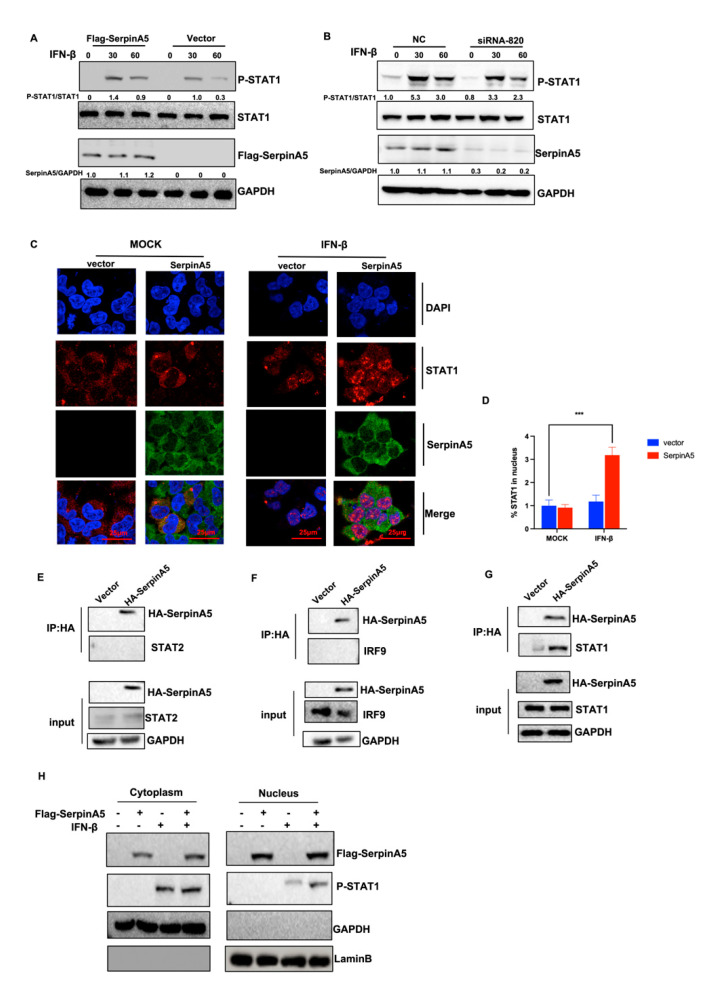
SerpinA5 promoted STAT1 phosphorylation and nuclear translocation. (**A**) Western blotting analysis of the expression level of STAT1 and phosphorylated STAT1 (P-STAT1) in vector or SerpinA5 transfected A549 cells with IFN-β stimulation (100 ng/mL). (**B**) Western blotting analysis of the expression level of STAT1and P-STAT1 in negative control (NC) or SerpinA5-knockdown THP-1 cells with IFN-β stimulation (100 ng/mL). (**C**) HEK-293T cells were transfected with 2 μg of Flag–SerpinA5–expressing plasmids for 24 h. The cells were then mock-infected or stimulated by IFN-β for 30 min and subjected to confocal microscopy analysis using the anti-Flag and anti-STAT1 antibodies. The scale bar represents 25μm. These cells shown in visual field represent three independent experiments. (**D**) Quantification of the percentage of STAT1 in the transfected vector or Flag–SerpinA5–expressing plasmids cells in the nucleus with or without IFN-β treatment. (**E**–**G**) HEK293T cells were transfected with HA-SerpinA5 or empty vector and 36 h later the cell lysates were collected and subjected to co-IP analysis using an anti-HA tag antibody. The whole-cell lysates and IP complexes were analyzed by Western blotting using antibodies against STAT1, STAT2, IRF9, HA, and GAPDH. (**H**) HEK293T cells were transfected with Flag-SerpinA5 or empty vector for 24 h, and then treated with IFN-β (100 ng/mL) for 6 h. The nuclear and cytoplasmic fractions were isolated and subjected to Western blot analysis with antibodies against p-STAT1 and Flag-SerpinA5. Data from at least triplicates were shown as the mean ± SD. *** *p* < 0.001.

**Figure 5 ijms-24-05458-f005:**
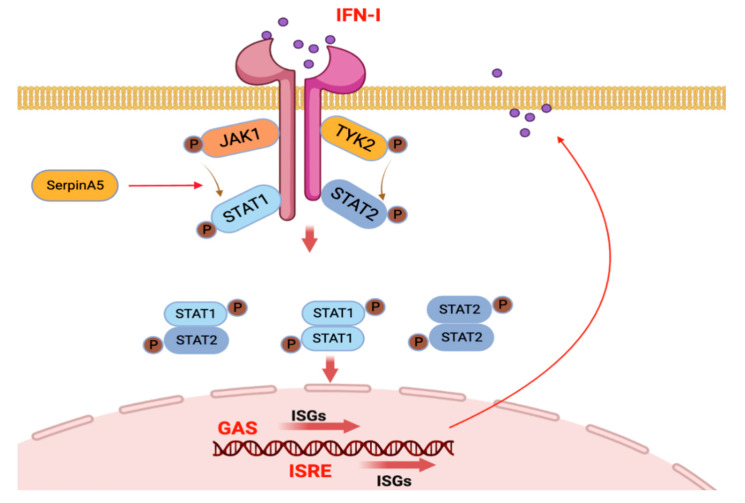
Pattern to illustrate the SerpinA5-involved innate immune signaling in response to viral infections. SerpinA5 can interact with STAT1 and upregulate the phosphorylation of STAT1 by promoting the recruitment of JAK1 kinase, and thus facilitate the formation of STAT1/STAT2 complex and its nuclear translation to bind to the ISRE (interferon-stimulated response element) and GAS (γ-activated sequence) sites. Subsequently, the transcription of IFN-stimulated genes (ISG) is initiated, and IFN-related signaling pathways are effectively activated to impair viral infections. This figure was created by BioRender.com.

## Data Availability

The data presented in this study are available in the main text and Appendix A.

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
