# Peer review of "IFN-Inducible SerpinA5 Triggers Antiviral Immunity by Regulating STAT1 Phosphorylation and Nuclear Translocation"

_ijms, 2023, doi:10.3390/ijms24065458_

Round 1
Reviewer 1 Report (Previous Reviewer 2)
No comments. Ready for publication
Author Response
Thank you for your kind approve our work.
Reviewer 2 Report (Previous Reviewer 4)
1. Correct the symbols (-+) of figure 1F since they do not coincide with the well
2. In the results section, some cell lines and transfection times are described, while others do not, as in section 2.2, which only mention "as described in Methods", the ideal would be to put it as they do in section 2.4, where they describe cell line, what technique did they find seems better to me that way.
3. In the legend of figure 2 there is no homogeneity, in some experiments they describe cell line and transfection time and in others they do not, as for example in figure A they do not say time or in which cell line, unlike figure I-M which does. do. They must homogenize in all descriptions of the figure legend.
4. Improve the quality of the figures
5. Line 267 wrote serpina5 is all in lowercase (they always write it as SerpinA5) review throughout the writing.
6. In fig 4A and 4B they mention that the overexpression of SarpinA5 increases P-STAT in the presence of SerpinA5, however in the Wb it looks the same as with the vector, however the authors put the density data, they would have another Wb where Is the effect even more visible?
7. Figure 4D review the legend of the graph the sign of % goes to the beginning (% STAT in nucle)
8. In the figure of the IF it is P-STAT or STAT TOTAL ?? If it is P-STAT, correct the figure.
9. In figure 4H, how do you explain that there are more pSTAT in the cytoplasm than in the nucleus?
10. In section 2.4 the description of the following result "In addition, changes in STAT1 phosphorylation and nuclear translocation in response to SerpinA5 stimulation were also confirmed at the protein level by Western blotting analysis (Fig. 4H). Mention that it was by cell fractionation.
11. Line 393 there is a double dot (..)
12. The discussion section feels like a summary of the results with an introduction and very little discussion of the data.
For example in Fig 2G it seems that SerpinA5 can cooperate to the expression of INF beta this by luciferase assays? So it could favor the expression of IFN and also of ISGs? Or how could you explain that?
Author Response
- Correct the symbols (-+) of figure 1F since they do not coincide with the well.
Answer: Thank you for your mention. We have corrected it.
2. In the results section, some cell lines and transfection times are described, while others do not, as in section 2.2, which only mention "as described in Methods", the ideal would be to put it as they do in section 2.4, where they describe cell line, what technique did they find seems better to me that way.
Answer: Thank you for your careful reading. We have added these descriptions accordingly.
- In the legend of figure 2 there is no homogeneity, in some experiments they describe cell line and transfection time and in others they do not, as for example in figure A they do not say time or in which cell line, unlike figure I-M which does. do. They must homogenize in all descriptions of the figure legend.
Answer: Thank you for your mention. We have added these descriptions accordingly.
- Improve the quality of the figures.
Answer: Thank you for your mention. We have provided the revised figures with high quality in the revised manuscript.
- Line 267 wrote serpina5 is all in lowercase (they always write it as SerpinA5) review throughout the writing.
Answer: Thank you for your careful reading. We have corrected this typo.
- In fig 4A and 4B they mention that the overexpression of SarpinA5 increases P-STAT in the presence of SerpinA5, however in the Wb it looks the same as with the vector, however the authors put the density data, they would have another Wb where Is the effect even more visible?
Answer: Thank you for your kind reminder. We have repeated this experiment and changed this figure in the revised manuscript.
- Figure 4D review the legend of the graph the sign of % goes to the beginning (% STAT in nucle)
Answer: Thank you for your kind mention. We have corrected it.
- In the figure of the IF it is P-STAT or STAT TOTAL?? If it is P-STAT, correct the figure.
Answer: Thank you for your kind mention. STAT1 was mainly distributed in the cytoplasm before stimulation, and we found that SerpinA5 overexpression significantly promoted STAT1 nuclear translocation by immunofluorescence assay (Fig.4C). In this experiment, we used the fluorescence-labled anti-STAT1 antibody, but not anti- P-STAT1 antibody, to reflect the STAT1 nucleus translocation. Thank you.
- In figure 4H, how do you explain that there are more pSTAT in the cytoplasm than in the nucleus?
Answer: Thank you for your careful reading. In figure 4H, Western blotting assay was performed for these two images (left-cytoplasm, right-nucleus) respectively, so these specific protein bands might be visualized in different exposure time and thus the color darkness was also different. As a result, we don’t think it can be compared between these two images. Nevertheless, this figure can show the difference of pSTAT1 amount in cytoplasm or in nucleus alone. Thank you.
- In section 2.4 the description of the following result "In addition, changes in STAT1 phosphorylation and nuclear translocation in response to SerpinA5 stimulation were also confirmed at the protein level by Western blotting analysis (Fig. 4H). Mention that it was by cell fractionation.
Answer: Thank you for your mention. We have added these descriptions accordingly in the revised section 2.4.
- Line 393 there is a double dot (..)
Answer: Thank you for your mention. We carefully checked and corrected these typos.
- The discussion section feels like a summary of the results with an introduction and very little discussion of the data. For example in Fig 2G it seems that SerpinA5 can cooperate to the expression of INF beta this by luciferase assays? So it could favor the expression of IFN and also of ISGs? Or how could you explain that?
Answer: Thank you for your kind suggestion. We have accordingly added more discussion about our results in the revised manuscript. For example, “we found that Serpin5 was a positive regulator in activating the IFN-related signaling pathways. For example, SerpinA5 treatment can upregulate the expression of IFN-related genes including IFN-β, MX1, CXCL10, IL-1β, IFN-λ. Mechanistically, like other ISGS (RIG-1, MDA-5, STATs, and TAP1 etc), SerpinA5 can regulate the PRRs, IFN-a/-β, and JAK/STAT together to form a positive feedback loop to promote the IFN production. Consequently, the secreted IFNs bind to type I IFN receptors (IFNAR1 and IFNAR2) and activate JAK family kinases to regulate the expression of more ISGs.”
Altogether, all of the above comments and suggestions are very helpful in improving our manuscript. We have taken these comments into account seriously and try our best to address these questions point-by-point. Thank you again for all of your comments and suggestions.

Round 2
Reviewer 2 Report (Previous Reviewer 4)
The paper improved
In the legend of figure 2 A, indicate the transfection time (24 hours).
In the material and methods section:
Confocal Microscopy
They do not mention which antibodies like they do in Wb and CoIP
Why don't I use p-stat in immunofluorescence?
Author Response
1.In the legend of figure 2 A, indicate the transfection time (24 hours).
Answer: Thank you for your careful reading. We have added this information in the revised manuscript.
2.In the material and methods section: Confocal Microscopy
They do not mention which antibodies like they do in Wb and CoIP
Answer: Thank you for your kind mention. We have added this information in the section of confocal microscopy assay.
3.Why don't I use p-stat in immunofluorescence?
Answer: Thank you for your kind mention. The binding of type I IFN induces the formation of a receptor complex between IFNAR1 and IFNAR2, leading to the activation of the receptor associated JAK1 and TYK2 kinases. This is followed by the recruitment and phosphorylation of STAT1 in the cytoplasm, and then the P-STAT1 can be translocated into nuclear. In order to reflect the total signal dynamic change of both the cytoplasm STAT1 and the P-STAT1 nuclear translocation, it usually uses the STAT antibody for this experiment. Another reason, the process of phosphorylation is usually very fast. Although the phosphatase inhibitors can be used in making phosphorylation samples for WB assay, it is difficult for confocal samples which often take a long process with in situ detection of living cells, so many studies usually use the STAT antibody for this experiment. For example, one recent study reported that poxviruses and paramyxoviruses use a conserved mechanism of STAT1 antagonism to inhibit interferon signaling, and they used the STAT1 antibody to confirm the ability of a poxvirus protein to inhibit STAT nuclear translocation[1]. Thank you again for your kind comments and suggestions.
References
- Talbot-Cooper, C., et al., Poxviruses and paramyxoviruses use a conserved mechanism of STAT1 antagonism to inhibit interferon signaling. Cell Host Microbe, 2022. 30(3): p. 357-372 e11.

This manuscript is a resubmission of an earlier submission. The following is a list of the peer review reports and author responses from that submission.
Round 1
Reviewer 1 Report
Reviewer’s comments
Manuscript ID: ijms-2056513
Title: IFN-inducible SerpinA5 triggers antiviral immunity by regulating STAT1
phosphorylation and nuclear translocation
Dear Editor
Thank you for inviting me to review the manuscript submitted to the IJMS Journal. But I regret to inform you that I’d suggest rejecting the current version of the manuscript. I provided comments and suggestions herein.
General comments:
The authors primarily found that activation of PRRs by viral mimic PAMPs (Poly I:C and R848), IFN-I agonist (LPS), and HSV-1 infection could upregulate expression of SerpinA5 mRNA and protein in BMM cells. Then, by employing IFNAR deficient BMM, it was demonstrated that SerpinA5 expression was dependent on IFN-I activation. Moreover, authors showed that SerpinA5 elicited anti-HSV-1 activity in HeLa, A549, and HEK293T cells. Hence, authors proposed that SerpinA5 was a novel ISG. Subsequently, authors dissected antiviral mechanisms of SerpinA5 by utilizing RNA-Seq, transcriptomic analysis, pathway identification by Western blot analysis and nuclear translocation assay to detect activated STAT1. However, the research design is not appropriate which seems not to have sound scientific merit. Moreover, the overall quality of figures and captions, as well as introduction and discussion, seem to be inappropriate for publication.
Specific comments:
Major comments:
1. To clarify underlying mechanisms of antiviral activity of SerpinA5, authors transfected A549 cells with plasmid expressing SerpinA5 and infected with HSV-1. Infected-cells transfected with empty plasmid served as control. This methodology seems to be inappropriate. The infection of A549 cells with HSV-1 can stimulate IFN-I production which turn to act as autocrine and paracrine to stimulate expression of various ISGs through JAK/STAT pathways. Hence, the effects of SerpinA5 on antivirus activity in A549 cells seem to be overwhelmed by the HSV-1 infection. The direct effect of SerpinA5 cannot be elucidated by the methodology done in this study.
2. For cGAS-STING signaling pathway, it has been well-characterized that cGAS is dsDNA cytoplasmic sensor which getting activation upon dsDNA engagement. The activated cGAS then synthesize 2′3′ cyclic GMP–AMP (cGAMP). The cGAMP binds to stimulator of interferon genes (STING) dimers which leading to recruitment of TANK-binding kinase 1 (TBK1) to initiate downstream signaling resulting in expression of IFN-I. In this study, it did not make any sense to detect SerpinA5-cGAS interaction. Moreover, the co-transfection of plasmids expressing cGAS, STING, TBK1, IRF3, and IRF7 with plasmids expressing SerpinA5 could not exclusively indicate effect of SerpinA5 on IFN-I induction.
3. Even authors had concluded SerpinA5 was ISG and mentioned that they aimed to clarify underlying mechanisms of antiviral activity of SerpinA5, in the subsequent experiments, they tried to demonstrated induction of IFN and activation of JAK/STAT signal transduction by SerpinA5. It is wired.
4. It would be helpful for authors to carefully revisit literature reviews on IFN-I pathways.
Minor comments:
There are too much typo errors and inconsistency in the main text. Please re-check thoroughly.
1. Lines 86 and 111, please check siRNA to SerpinA5 that really used (siRNA329 or siRNA820).
2. Figure 1:
2.1 panels G and J, should indicate amounts of plasmid over the figure
2.2 panel I, there are 2 lines, why?
2.3 panels I and L, are error bars missing?
2.4 panels H and K, what is x-axis, amounts of plasmid?
2.5 panel N, treatments should be indicated over the figure
3. Line 100, should “intern control” be “internal control”?
4. Line 104, should “J” be “(J)”?
5. Line 105, please check “…..0, 0.5, 1, or 1 lg….”
6. Line 130, should “INF-stimulated response element” be “IFN-stimulated response element”?
7. Lines 132 and 151, please check consistency of genes that determined by RT-qPCR (IFN-b, MX1, CXCL10, IL-1b, and IFN-l) mentioned in text with the legend of Fig. 2 (IFN-b, ISG15, PKR, MX1 , and IL-1b).
8. Figure 2:
8.1 panel C, too tiny lists of gene
8.2 repeat label panel H
8.3 panels G and H, please mention amount of plasmid (in line 146, figure legend, mentioned 3 doses, please re-check)
9. In Section 2.2, please clarify the rationale for selecting genes to be verified by RT-qPCR. Are they identified by RNA-Seq and found in transcriptomes?
10. Lines 185, should “3.4” be “2.4”?
11. Line 159, Ref. 18 seems to be inappropriate citation, please recheck.
12. In Materials and Methods, line 296, please give nucleotide sequences of siRNAs.
13. In Materials and Methods, line 314, S1 table could not be found.
14. In Materials and Methods, line 325, should “Pathway function analysis” be “pathway function analysis”?
Best regards

Author Response
General comments: The authors primarily found that activation of PRRs by viral mimic PAMPs (Poly I:C and R848), IFN-I agonist (LPS), and HSV-1 infection could upregulate expression of SerpinA5 mRNA and protein in BMM cells. Then, by employing IFNAR deficient BMM, it was demonstrated that SerpinA5 expression was dependent on IFN-I activation. Moreover, authors showed that SerpinA5 elicited anti-HSV-1 activity in HeLa, A549, and HEK293T cells. Hence, authors proposed that SerpinA5 was a novel ISG. Subsequently, authors dissected antiviral mechanisms of SerpinA5 by utilizing RNA-Seq, transcriptomic analysis, pathway identification by Western blot analysis and nuclear translocation assay to detect activated STAT1. However, the research design is not appropriate which seems not to have sound scientific merit. Moreover, the overall quality of figures and captions, as well as introduction and discussion, seem to be inappropriate for publication.
Answer: Thank you very much for your critical comments and suggestions, which are very helpful in improving our manuscript. We have taken these comments into account seriously and try our best to address these questions point-by-point. After making extensive revisions, we now submit this revised manuscript, and we believe this version has been improved substantially. Thank you again for all of your comments and suggestions.
Specific comments:
Major comments:
- To clarify underlying mechanisms of antiviral activity of SerpinA5, authors transfected A549 cells with plasmid expressing SerpinA5 and infected with HSV-1. Infected-cells transfected with empty plasmid served as control. This methodology seems to be inappropriate. The infection of A549 cells with HSV-1 can stimulate IFN-I production which turn to act as autocrine and paracrine to stimulate expression of various ISGs through JAK/STAT pathways. Hence, the effects of SerpinA5 on antivirus activity in A549 cells seem to be overwhelmed by the HSV-1 infection. The direct effect of SerpinA5 cannot be elucidated by the methodology done in this study.
Answer: Thank you for your question. In this study, we firstly investigated whether SerpinA5 could exert an unrecognized role in antiviral innate immunity. As shown in Figure 1, Hela cells and A549 cells were transfected with SerpinA5-expressing plasmids for 24 h, and then infected with HSV-1. Results demonstrated that SerpinA5 overexpression in Hela cells significantly suppressed HSV-1 replication (Fig. 1G, 1H). The direct inhibition of viral replication by SerpinA5 was further confirmed by TCID50 assay (Fig. 1I). The similar results were also observed in SerpinA5-treated A549 cells (Fig. 1J-L).
To further verify the direct inhibitory effect of SerpinA5 on viral replication, we designed the siRNA to downregulate the expression of SerpinA5 (Fig.1M and 1N), and these SerpinA5-knockdown-cells became more susceptible to HSV-1 infection when compared to that mock-treated cells (Fig.1O and 1P). Together, these results indicated that SerpinA5 played an direct antiviral function.
Then, to further clarify the underlying mechanism for this antiviral activity, we analyzed which innate immunity-related genes were altered in response to SerpinA5 treatment during HSV-1 infection. As you mentioned, HSV-1 infection can stimulate IFN-I pathway in some extent , but we found that the activation of IFN-I pathway can be further increased in response to SerpinA5 treatment during HSV-1 infection. Thus, we speculated that SerpinA5 can play the antiviral activity by regulating IFN-I pathway. Moreover, this finding was further confirmed by the promoter luciferase reporter system, and we found that SerpinA5 overexpression with or without HSV-1 infection significantly promoted the activation of IFN-β promoter and ISRE promoter (Fig. 2G-H). Taken together, these results suggested that SerpinA5 played the antiviral function through modulating IFN-related signaling pathways.
- For cGAS-STING signaling pathway, it has been well-characterized that cGAS is dsDNA cytoplasmic sensor which getting activation upon dsDNA engagement. The activated cGAS then synthesize 2′3′ cyclic GMP–AMP (cGAMP). The cGAMP binds to stimulator of interferon genes (STING) dimers which leading to recruitment of TANK-binding kinase 1 (TBK1) to initiate downstream signaling resulting in expression of IFN-I. In this study, it did not make any sense to detect SerpinA5-cGAS interaction. Moreover, the co-transfection of plasmids expressing cGAS, STING, TBK1, IRF3, and IRF7 with plasmids expressing SerpinA5 could not exclusively indicate effect of SerpinA5 on IFN-I induction.
Answer: As mentioned above, our results suggested that SerpinA5 might play the antiviral function through modulating IFN-related signaling pathways (Figure 2). Next, we want to further explore how SerpinA5 activated the IFN production in exact molecular mechanism, including cGAS-STING, JAK-STAT, RIG-I, TLR and other pathways. Our previous work (Zhao et al. Plos Pathogens, 2022) demonstrated that SerpinA5 can be induced in response to HSV-1 stimulation, which is a classical dsDNA virus model to activate the cGAS-STING signaling pathway. Accordingly, we speculated whether SerpinA5 might play the antiviral activity by regulating the cGAS-STING pathway. Thus, we initially performed a series experiment to verify or exclude this hypothesis (Figure 3).
- Even authors had concluded SerpinA5 was ISG and mentioned that they aimed to clarify underlying mechanisms of antiviral activity of SerpinA5, in the subsequent experiments, they tried to demonstrated induction of IFN and activation of JAK/STAT signal transduction by SerpinA5. It is wired.
Answer:As mentioned above, after we showed that SerpinA5 can activate the IFN production independent of cGAS-STING pathway, we then verified that SerpinA5 can upregulate the phosphorylation of STAT1 and promote its nuclear translocation, thus effectively activating the transcription of IFN-related signaling pathways to impair viral infections.
- It would be helpful for authors to carefully revisit literature reviews on IFN-I pathways.
Answer: Thank you for your kind mention. We have extensively modified this manuscript accordingly.
Minor comments:
There are too much typo errors and inconsistency in the main text. Please re-check thoroughly.
- Lines 86 and 111, please check siRNA to SerpinA5 that really used (siRNA329 or siRNA820).
Answer: Thank you for your mention. We have revised it carefully, the siRNA820 was used in the knockdown experiment.
- Figure 1:
2.1 panels G and J, should indicate amounts of plasmid over the figure
Answer: Thank you for your mention. The transfection doses of pCDNA3.1-Flag-SerpinA5 in each well was 0, 0.5, 1 or 2ug respectively, and we also simultaneously transfected pCDNA3.1 vector to maintain the total 2ug of plasmids in each well.
2.2 panel I, there are 2 lines, why?
Answer: Two lines represented the results of repeated experiments. We have repeated this experiment with three times in revised Figure 1.
2.3 panels I and L, are error bars missing?
Answer: We have revised Figure 1 accordingly as mentioned above.
2.4 panels H and K, what is x-axis, amounts of plasmid?
Answer: Yes, x-axis is the transfection doses of pCDNA3.1-Flag-SerpinA5 in each well, and we have provided this information in revised Figure legends.
2.5 panel N, treatments should be indicated over the figure
Answer: The treatment for panel M and panel N is same, and we have descripted this information in the revised Figure legend. Thank you.
- Line 100, should “intern control” be “internal control”?
Answer: Thank you for your careful reading. We have corrected it.
- Line 104, should “J” be “(J)”?
Answer: Thank you for your careful reading. We have corrected it.
- Line 105, please check “…..0, 0.5, 1, or 1 lg….”
Answer: Thank you for your careful reading. We have corrected it.
- Line 130, should “INF-stimulated response element” be “IFN-stimulated response element”?
Answer: Thank you for your careful reading. We have corrected it.
- Lines 132 and 151, please check consistency of genes that determined by RT-qPCR (IFN-b, MX1, CXCL10, IL-1b, and IFN-l) mentioned in text with the legend of Fig. 2 (IFN-b, ISG15, PKR, MX1 , and IL-1b).
Answer: Thank you for your careful reading. We have corrected it.
- Figure 2:
8.1 panel C, too tiny lists of gene
Answer: Thank you for your mention. We have redrawn this figure.
8.2 repeat label panel H
Answer: Thank you for your careful reading. We have corrected it.
8.3 panels G and H, please mention amount of plasmid (in line 146, figure legend, mentioned 3 doses, please re-check)
Answer: Thank you for your careful reading. We have corrected it.
- In Section 2.2, please clarify the rationale for selecting genes to be verified by RT-qPCR. Are they identified by RNA-Seq and found in transcriptomes?
Answer: To clarify the underlying mechanism for the antiviral activity of SerpinA5, we subsequently performed the RNA-Seq analysis, and our results implied that SerpinA5 might play the antiviral function through modulating IFN-related signaling pathways. Therefore, we further confirmed some IFN-related genes by RT-qPCR, including IFN-β, MX1, CXCL10, IL-1β, IFN-λ.
- Lines 185, should “3.4” be “2.4”?
Answer: Thank you for your careful reading. We have corrected it.
- Line 159, Ref. 18 seems to be inappropriate citation, please recheck.
Answer: Thank you for your careful reading. We have corrected this Ref accordingly in the revised manuscript.
- In Materials and Methods, line 296, please give nucleotide sequences of siRNAs.
Answer: Thank you for your careful reading. We have provided it in the revised manuscript.
- In Materials and Methods, line 314, S1 table could not be found.
Answer: Thank you for your careful reading. We have provided it in the revised manuscript.
- In Materials and Methods, line 325, should “Pathwayfunction analysis” be “pathwayfunction analysis”?
Answer: Thank you for your careful reading. We have corrected it accordingly.
Reviewer 2 Report
The authors have completed a thorough study upon the participation of SerpinA5 to IFN response pathway. They use virus induction or others inducers of ISGs in order to prove the involvement of SerpinA5 to their expression. They also utilize overexpression and silencing technologies to prove the importance of this molecule to the full activation and propagation of type I IFN signaling. Though the experimental plan is thoroughly designed, the interpretation of the results remains pure and the manuscript at its current form does not meet publishing criteria.
Comments
1. Line 19 nuclear translocation, not translation
2. Line 44 "is a family member of glycoproteins"
3. Line 46, role, not roles
4. Line 47 "antimicrobe what??"
5. Line 52 yet is redundant
6. Line 55-56 "This comprehensive study will contribute to the clarification of the biological role of SerpinA5, by deeper understanding....
7. Results 2.1 Where are the results of inducing the other cell lines? RAW and THP-1
8. Line 70 is instead of was. In general the authors switch often between past and present in verbs describing their results, which is painful for the reader.
9. In the main text, in 2.1 results there is no mention of HEK293 cells as the ones that kd of serpina5 is taking place...
10. Line 74 respectively is redundant.
11. Results 2.2 rephrase the tense of title
12. Lines 116-119. What experimental setup (cells, trigger, technology etc) has been used in the "unbiased transcriptomic analysis"? These details are fully omitted in the main text. They appear only in Materials and Methods.
13. In the legend of figure 2 it is written "Data from at least triplicate experiments were shown as the mean". What is exactly the case for each experiment? In the main text, authors write that the transciptional analysis was performed in two independent experiments. Is this two biological replicates? or two sets of biological replicates? This is not apparent in the figure of the results, too.
14. The authors should explain in general the utilization of the ISRE containing construct, as this region is generic for ISGs and its specific significance depends on the context.
15 Lines 119-126 should be merged
16. What are the DEGs? is there any list deposited? Any deposition of the data in a biobank?
17. The results of the RNA-seq experiment are poorly discussed.
18. In general, technical details of the experimental setup (heavy details as the ng of plasmid transfected and so on) should be moved to materials and methods. Please make legends shorter and more "reader-friendly"
19. Lines 164-167. Inconclusive conclusion. The authors have detected that cGAS-STING components upon overexpression increase ISREmediated luciferase expression, and on top of that SerpinA5 o/e superactivates this system. Then they conclude that " indicating that SerpinA5 might activate IFN production by targeting the downstream of IRF3 and IRF7". Not clear how this is true...Probably on the contrary...
20. Line 168. not to further exclude, rather to test.
21. Lines 171-172. How the absence of the direct interaction of serpinA5 with the key molecules of cGAS-STING pathway proves that Serpin5A acts independently of them in order to activate IFN pathway? The authors have just proved that cGAS-STING components along with Serpin5a superactivate the pathway, and then they jump to this conclusion. In case their aim was to prove such and independency, they should have tried complete blockage of the cGAS-STING signaling branch to check whether serpin5a can still activate IFN signaling...
22. Results 3.4 The experiments start from a wrong conclusion "downstream of IRF3 and IRF7". What exactly is downstream of this molecules? the STAT1 connection is unclear. Nevertheless, after the authors prove that o/e of serpinA5 induces phosphorylation of STAT1 and consequently translocation in the nucleus (and only that!!), they postulate different findings in the discussion (see comments below). They also see that kd of serpinA5 brings the opposite effect.
23. The findings discussed in comment 22 solely show again the activation of IFN signaling upon serpin5A o/e and its importance in this pathway. In the discussion the authors in two points they say: 1. we also provided evidence that SerpinA5 can upregulate the phosphorylation of STAT1 by promoting the recruitment of JAK1 kinase (bold"not true) and then 2. Based on our results, we speculated that SerpinA5 may promote TYK2 recruitment to STAT1, and function as a scaffold to stabilize the SH2 domain. Why is that? The authors here jump into assumptions not scientifically built. Either rephrase or remove...
24. Findings of figure are not depicted in the model of figure 5.
25. Discussion is pure, focused mainly on the findings of figure 4 which is misleading as these results are overvalued. Authors should elaborate on the findings of their transcriptomic study.
26. In another comment, since serpinA5 is transferred into the nucleus and the authors have the opportunity to run co-IP assays, a good idea might be to check serpinA5-stat direct interaction as a facilitator of ISG expression.
27. What is the software for the quantification of the confocal experiments?
Author Response
The authors have completed a thorough study upon the participation of SerpinA5 to IFN response pathway. They use virus induction or others inducers of ISGs in order to prove the involvement of SerpinA5 to their expression. They also utilize overexpression and silencing technologies to prove the importance of this molecule to the full activation and propagation of type I IFN signaling. Though the experimental plan is thoroughly designed, the interpretation of the results remains pure and the manuscript at its current form does not meet publishing criteria.
Answer: Thank you for your comments. We have taken these comments into account seriously and try our best to address these questions point-by-point. After making extensive revisions, we now submit this revised manuscript, and we believe this version has been improved substantially. Thank you again for all of your comments and suggestions.
1.Line 19 nuclear translocation, not translation
Answer: Thank you for your careful reading. We have corrected it accordingly.
2.Line 44 "is a family member of glycoproteins"
Answer: Thank you for your careful reading. We have corrected it accordingly.
3.Line 46, role, not roles
Answer: Thank you for your careful reading. We have corrected it accordingly.
4.Line 47 "antimicrobe what??"
Answer: Thank you for your careful reading. We have corrected it accordingly.
5.Line 52 yet is redundant
Answer: Thank you for your careful reading. We have corrected it accordingly.
6.Line 55-56 "This comprehensive study will contribute to the clarification of the biological role of SerpinA5, by deeper understanding....
Answer: Thank you for your careful reading. We have corrected it accordingly.
7.Results 2.1 Where are the results of inducing the other cell lines? RAW and THP-1
Answer: Thank you for your careful reading. We have provided these results in the Supplementary Fig.1.
8.Line 70 is instead of was. In general the authors switch often between past and present in verbs describing their results, which is painful for the reader.
Answer: Thank you for your careful reading. We have corrected it accordingly.
9.In the main text, in 2.1 results there is no mention of HEK293 cells as the ones that kd of serpina5 is taking place...
Answer: Thank you for your kind reminder. We have mentioned it in revised manuscript.
10.Line 74 respectively is redundant.
Answer: Thank you for your careful reading. We have corrected it accordingly.
11.Results 2.2 rephrase the tense of title
Answer: Thank you for your careful reading. We have corrected it accordingly.
12.Lines 116-119. What experimental setup (cells, trigger, technology etc) has been used in the "unbiased transcriptomic analysis"? These details are fully omitted in the main text. They appear only in Materials and Methods.
Answer: Thank you for your careful reading. We have corrected it accordingly.
13.In the legend of figure 2 it is written "Data from at least triplicate experiments were shown as the mean". What is exactly the case for each experiment? In the main text, authors write that the transciptional analysis was performed in two independent experiments. Is this two biological replicates ? or two sets of biological replicates? This is not apparent in the figure of the results, too.
Answer: Thank you for your kind reminders. Data (Figure 2 G-N) from at least triplicate experiments were shown as the mean ± SD. As for the two independent experiments of transcriptome analysis, it meant two times of RNAseq test with different samples. Figure 2 represented the RNAseq result of one independent experiment. We have stated this information clearly in the revised manuscript.
14.The authors should explain in general the utilization of the ISRE containing construct, as this region is generic for ISGs and its specific significance depends on the context.
Answer: Thank you for your mention. Based on above transcriptome data, we focused that SerpinA5 might play the antiviral function through modulating IFN signaling pathways. Therefore, we want to further confirm this finding by the luciferase reporter system of IFN-β promoter and ISRE promoter.
15.Lines 119-126 should be merged
Answer: Thank you for your careful reading. We have adjusted it accordingly.
16.What are the DEGs? is there any list deposited? Any deposition of the data in a biobank?
Answer: We have added a detailed explanation of differentially expressed genes (DEGs) in the methods section. “According to their expression levels in different sample groups, these genes with Fold Change≥2 and FDR<0.05 were considered differentially expressed genes (DEGs). Fold Change represents the ratio of expression between two samples (groups). The False Discovery Rate (FDR) is obtained by correcting the p-value of the difference significance using the recognized Benjamini-Hochberg correction method.” We also provided the full set of DEGs in the supplement material.
17.The results of the RNA-seq experiment are poorly discussed.
Answer: We have provided more discussion about RNA-seq data in the revised manuscirpt. (Line274-278)
18.In general, technical details of the experimental setup (heavy details as the ng of plasmid transfected and so on) should be moved to materials and methods. Please make legends shorter and more "reader-friendly"
Answer: Thank you for your kind reminders. We have adjusted these details accordingly in the revised manuscript.
19.Lines 164-167. Inconclusive conclusion. The authors have detected that cGAS-STING components upon overexpression increase ISRE mediated luciferase expression, and on top of that SerpinA5 o/e superactivates this system. Then they conclude that " indicating that SerpinA5 might activate IFN production by targeting the downstream of IRF3 and IRF7". Not clear how this is true...Probably on the contrary...
Answer: As mentioned above, our results suggested that SerpinA5 might play the antiviral function through modulating IFN-related signaling pathways (Figure 2). Next, we want to further explore how SerpinA5 activated the IFN production in exact molecular mechanism, including cGAS-STING, JAK-STAT, RIG-I, TLR and other pathways. Our previous work (Zhao et al. Plos Pathogens, 2022) demonstrated that SerpinA5 can be induced in response to HSV-1 stimulation, which is a classical dsDNA virus model to activate the cGAS-STING signaling pathway. Accordingly, we speculated whether SerpinA5 might play the antiviral activity by regulating the cGAS-STING pathway. Thus, we initially performed a series experiment to verify or exclude this hypothesis (Figure 3). After we showed that SerpinA5 can activate the IFN production independent of cGAS-STING pathway, we then verified that SerpinA5 can upregulate the phosphorylation of STAT1 and promote its nuclear translocation, thus effectively activating the transcription of IFN-related signaling pathways to impair viral infections. In addition, since the luciferase reporter system of ISRE promoter can be activated both in cGAS-STING pathway and JAK-STAT pathway, we don’t think this is a contrary result. Thank you.
20.Line 168. not to further exclude, rather to test.
Answer: Thank you for your careful reading. We have corrected it accordingly.
21.Lines 171-172. How the absence of the direct interaction of serpinA5 with the key molecules of cGAS-STING pathway proves that Serpin5A acts independently of them in order to activate IFN pathway? The authors have just proved that cGAS-STING components along with Serpin5a superactivate the pathway, and then they jump to this conclusion. In case their aim was to prove such and independency, they should have tried complete blockage of the cGAS-STING signaling branch to check whether serpin5a can still activate IFN signaling..
Answer: Thank you for your nice suggestion. We blocked the cGAS-STING signaling using C-176, which is a kind of STING inhibitor. Results showed that SerpinA5 overexpression can upregulate the expression of IFN-β and ISG56 even in presence of C-176 treatment(Fig.3G).
22.Results 3.4 The experiments start from a wrong conclusion "downstream of IRF3 and IRF7". What exactly is downstream of this molecules? the STAT1 connection is unclear. Nevertheless, after the authors prove that o/e of serpinA5 induces phosphorylation of STAT1 and consequently translocation in the nucleus (and only that!!), they postulate different findings in the discussion (see comments below). They also see that kd of serpinA5 brings the opposite effect.
Answer: Thank you for your kind suggestion. We have revised this sentence accordingly. “Except of cGAS-STING pathway, we further explored how SerpinA5 activated the IFN production in other molecular mechanisms, such as JAK-STAT pathway. Considering that STAT1 nuclear transcolation is an essential step to activate the downstream genes of IFN signaling cascades, we thus investigated whether SerpinA5 regulates JAK-STAT cascades.”
We also conducted additional experiments to verify that SerpinA5 can interact with STAT1 but can not interact with STAT2 and IRF9. Changes in STAT1 phosphorylation and nuclear translocation in response to SerpinA5 stimulation were also confirmed at the protein level by Western blotting analysis(Fig4E-4H).
- The findings discussed in comment 22 solely show again the activation of IFN signaling upon serpin5A o/e and its importance in this pathway. In the discussion the authors in two points they say: 1. we also provided evidence that SerpinA5 can upregulate the phosphorylation of STAT1 by promoting the recruitment of JAK1 kinase (bold"not true) and then 2. Based on our results, we speculated that SerpinA5 may promote TYK2 recruitment to STAT1, and function asEven authors had concluded SerpinA5 was ISG and mentioned that they aimed to clarify underlying mechanisms of antiviral activity of SerpinA5, in the subsequent experiments, they tried to demonstrated induction of IFN and activation of JAK/STAT signal transduction by SerpinA5. It is wired.
a scaffold to stabilize the SH2 domain. Why is that? The authors here jump into assumptions not scientifically built. Either rephrase or remove...
Answer: Thank you for your kind suggestions. We have adjust these descriptions accordingly in the revised manuscript.
- Findings of figure are not depicted in the model of figure 5.
Answer: We mentioned this model in the discussion section. “In summary, we identified that SerpinA5 could exert a previously unrecognized function as a novel ISG with antiviral activity, and we also proposed a working model for the roles of SerpinA5 in regulating innate immune signaling to control viral infection (Fig. 5).”
- Discussion is pure, focused mainly on the findings of figure 4 which is misleading as these results are overvalued. Authors should elaborate on the findings of their transcriptomic study.
Answer: Thank you for your kind suggestion. We have provided more discussion about RNA-seq data and the molecular mechanism involved in regulating IFN signaling pathway by SerpinA5 treatment in the revised manuscript.
- In another comment, since serpinA5 is transferred into the nucleus and the authors have the opportunity to run co-IP assays, a good idea might be to check serpinA5-stat direct interaction as a facilitator of ISG expression.
Answer: Thank you for your nice suggestion. We conducted the co-IP experiment to address this issue, and our results showed that serpinA5 can interact with STAT1 but can not interact with STAT2 and IRF9(Fig4F-4H)
- What is the software for the quantification of the confocal experiments?
Answer: The software image J was used for confocal experiments. We have provided this information in the revised manuscript. Thank you.
Reviewer 3 Report
The results are well presented and the study is interesting in the point of host-virus interactions. However minor language corrections are required before the formal publication in the journal.
Author Response
Thank you for your kind comments. The language has been carefully edited.
Reviewer 4 Report
1. In Fig1 they indicate that SerpinA5 increases after stimulation with different IFN inducers and is capable of inhibiting HSV1 infection, what could be the mechanism that SerpinA5 uses to inhibit viral infection?
2. in fig 1F the + - signs were moved.
3. Improve the quality of Fig 2
4. In the description of section “2.2. SerpinA5 played the antiviral function through modulating IFN signaling pathways” does not mention that They transfected and infected with HSV1 and then did RNA-seq, although it says in the figure legends, I think it is necessary to clarify it from the results.
5. Did you analyze which genes are altered only with the expression of SerpinA5?
*This could help to understand how SerpinA5 works against viral infections.
*Although RNAseq shows cellular processes associated with the IFN pathway, this could be because they are infected with HSV1.
6. On line 152 there is a double space
7. In Fig. 4A, They show that the overexpression of SerpinA5 affects the levels of PSTAT1, but in the Western Blot it is observed that both in FLAG-SerpinA5 and in the vector there is activation of P-STAT1, this may be due to the presence of IFNB1 and not by SerpinA5, this is also shown in fig 4C.
Therefore, the question would be if only SerpinA5 is really capable affect STAT?.
8. Homogenizing the figures, for example in Fig. 4B has the densitometry in western blotin both P-STAT1 and GAPDH, but in Fig. 4A it only has it in P-STAT1.
9. What is the effect of SerpinA5 overexpression on other ISGf3 components (STAT2 and IRF9).
10. How explain the effect observed in fig4A with respect to the fact that the presence of IFNB1 decreases the levels of SerpinA5.
11. In section “2.3. SerpinA5 activated IFN production independent of cGAS-STING signaling pathway”, could SerpinA5 activate other receptors capable of activating the IFN pathway such as TLRs?
12. The discussion is similar to the introduction and they discuss very little about their results.
Author Response
1.In Fig1 they indicate that SerpinA5 increases after stimulation with different IFN inducers and is capable of inhibiting HSV1 infection, what could be the mechanism that SerpinA5 uses to inhibit viral infection?
Answer: In this study, our data showed that SerpinA5, as a novel ISG, played a previously unrecognized antiviral activity. Mechanistically, SerpinA5 can upregulate the phosphorylation of STAT1 and promote its nuclear translocation, thus effectively activating the transcription of IFN-related signaling pathways to impair viral infections. SerpinA5-mediated innate immune signaling during the virus-host interactions.
In the description of section “2.2. SerpinA5 played the antiviral function through modulating IFN signaling pathways” does not mention that They transfected and infected with HSV1 and then did RNA-seq, although it says in the figure legends, I think it is necessary to clarify it from the results.
Answer: Thank you for your kind suggestion. We have mentioned it accordingly in revised manuscript.
2.in fig 1F the + - signs were moved.
Answer: Thank you for your kind reminder. We have adjusted it accordingly.
- Improve the quality of Fig 2
Answer: Thank you for your kind mention. We have redrawn this Figure.
4.Did you analyze which genes are altered only with the expression of SerpinA5?
*This could help to understand how SerpinA5 works against viral infections.
*Although RNAseq shows cellular processes associated with the IFN pathway, this could be because they are infected with HSV1.
Answer: Thank you for your suggestions. In this study, we firstly investigated whether SerpinA5 could exert an unrecognized role in antiviral innate immunity. As shown in Figure 1, Hela cells and A549 cells were transfected with SerpinA5-expressing plasmids for 24 h, and then infected with HSV-1. Results demonstrated that SerpinA5 overexpression significantly suppressed HSV-1 replication.To further verify the direct inhibitory effect of SerpinA5 on viral replication, we designed the siRNA to downregulate the expression of SerpinA5 (Fig.1M and 1N), and these SerpinA5-knockdown-cells became more susceptible to HSV-1 infection when compared to that mock-treated cells (Fig.1O and 1P). Together, these results indicated that SerpinA5 played an direct antiviral function.
Then, to further clarify the underlying mechanism for this antiviral activity, we analyzed which innate immunity-related genes were altered in response to SerpinA5 treatment during HSV-1 infection. As you mentioned, HSV-1 infection can stimulate IFN-I pathway in some extent , but we found that the activation of IFN-I pathway can be further increased in response to SerpinA5 treatment during HSV-1 infection. Thus, we speculated that SerpinA5 can play the antiviral activity by regulating IFN-I pathway. Moreover, this finding was further confirmed by the promoter luciferase reporter system, and we found that SerpinA5 overexpression with or without HSV-1 infection significantly promoted the activation of IFN-β promoter and ISRE promoter (Fig. 2G-H). Taken together, these results suggested that SerpinA5 played the antiviral function through modulating IFN-related signaling pathways.
- On line 152 there is a double space
Answer: Thank you for your careful reading. We have corrected it.
- In Fig. 4A, They show that the overexpression of SerpinA5 affects the levels of PSTAT1, but in the Western Blot it is observed that both in FLAG-SerpinA5 and in the vector there is activation of P-STAT1, this may be due to the presence of IFNB1 and not by SerpinA5, this is also shown in fig 4C.Therefore, the question would be if only SerpinA5 is really capable affect STAT?
Answer: Thank you for your question. In this experiment, THP-1 cells were transfected with or without SerpinA5-expressing plasmid, followed by IFN-β stimulation, and then the levels of STAT1 and p-STAT1 were quantified. As you mentioned, IFN-βcan activate the p-STAT1 in some extent , but we found that the activation of p-STAT1 can be further increased in response to SerpinA5 treatment during the presence of IFN-β(Fig.4A). SerpinA5 knockdown decreased the phosphorylation of STAT1 (Fig.4B). As well known, STAT1 was mainly distributed in the cytoplasm before IFN-β stimulation, and then transported into the nucleus after IFN-β stimulation. Interestingly, SerpinA5 overexpression significantly promoted STAT1 nuclear transportation (Fig.4C). Taken together, these results demonstrated that SerpinA5 can induce antiviral innate immunity by promoting STAT1 phosphorylation and nuclear translation.
7.Homogenizing the figures, for example in Fig. 4B has the densitometry in western blot in both P-STAT1 and GAPDH, but in Fig. 4A it only has it in P-STAT1.
Answer: Thank you for your kind remiders. We have revised all the Figures accordingly in the revised manuscript.
8.What is the effect of SerpinA5 overexpression on other ISGf3 components (STAT2 and IRF9).
Answer: Thank you for your question. We conducted IP experiment to address this issue, and our results showed that serpinA5 can interact with STAT1 but can not interact with STAT2 and IRF9(Fig4F-4H).
9.How explain the effect observed in fig4A with respect to the fact that the presence of IFNB1 decreases the levels of SerpinA5.
Answer: Thank you for your question. Actually, we don’t think IFN-β can affect the expression of Flag-SerpinA5. We transfected 293T cells with Flag-SerpinA5, and then treated them with IFN-β for 0h,0.5h,1h, and 6h. Results showed that that IFN-βdid not affect the expression of Flag-SerpinA5(S2).
10.In section “2.3. SerpinA5 activated IFN production independent of cGAS-STING signaling pathway”, could SerpinA5 activate other receptors capable of activating the IFN pathway such as TLRs?
Answer: Except of cGAS-STING pathway, we also explored how SerpinA5 activated the IFN production in other molecular mechanisms, such as JAK-STAT pathway. We found that SerpinA5 can upregulate the phosphorylation of STAT1 and promote its nuclear translocation, thus effectively activating the transcription of IFN-related signaling pathways to impair viral infections (Figure 4).
- The discussion is similar to the introduction and they discuss very little about their results.
Answer: Thank you for your reminder. We have rewritten this section accordingly in the revised manuscript.
Round 2
Reviewer 1 Report
Thank you for your revisions and clarifications. The role of SerpinA5 in the suppression of HSV replication has been clearly demonstrated. However, the underlying mechanism for antiviral activity has not been clearly elucidated yet since crucial controls are still missing in some experiments:
1. To validate the direct effect of SerpinA5 on modulating IFN-related signaling pathways, IFN-B induction, and STAT1 activation, authors should add more controls which are cells transfected with SerpinA5-expressing plasmids alone without either HSV-1 infection or IFN-B treatment and then detect the respective outcomes.
2. To exclusively conclude the independency of the cGAS-STING pathway on induction of IFN-B by SerpinA5, it'd be better if authors add more control which is SerpinA5-expressing plasmids alone without any treatment. And, the standard protocol for assessing activation of the cGAS-STING pathway i.e. Chamma H, Guha S, Laguette N, Vila IK. Protocol to induce and assess cGAS-STING pathway activation in vitro. STAR Protoc. 2022 May 14;3(2):101384. doi: 10.1016/j.xpro.2022.101384. should be employed.
Minor comments:
1. Figure 1H, 1I, 1K and 1L, x-axis should be plasmid?
2. Line 200: please recheck "serpina5".
Thank you
Author Response
Thank you for your revisions and clarifications. The role of SerpinA5 in the suppression of HSV replication has been clearly demonstrated. However, the underlying mechanism for antiviral activity has not been clearly elucidated yet since crucial controls are still missing in some experiments:
Answer: Thank you again for your kind comments and suggestions.
- To validate the direct effect of SerpinA5 on modulating IFN-related signaling pathways, IFN-B induction, and STAT1 activation, authors should add more controls which are cells transfected with SerpinA5-expressing plasmids alone without either HSV-1 infection or IFN-B treatment and then detect the respective outcomes.
Answer: In this study, we reported that SerpinA5 played a previously unrecognized antiviral activity, and then identified it effectively activated the IFN-related signaling pathway by a series of experiments, including RNA-seq, the promoter luciferase reporter system, RT-PCR, confocal microscopy analysis, and Co-IP etc (Figure 2, Figure 4). As for your mention that “authors should add more controls which are cells transfected with SerpinA5-expressing plasmids alone without either HSV-1 infection or IFN-B treatment”, We actually have shown this in the manuscript in Fig2.G-M. As described in our manuscript, A549 cells were transfected with SerpinA5–expressing plasmids for the luciferase reporter test of IFN-β promoter and ISRE promoter, and the cells were mock-infected (SerpinA5 plasmid alone, Blue column) or infected with HSV-1 (SerpinA5 plasmid + HSV-1, Red column). As expected, we found that SerpinA5 overexpression significantly promoted the activation of IFN-β promoter and ISRE promoter.
In addition, regarding to test the SerpinA5-involved virus-induced IFN signal pathway, some of our experimental design used “SerpinA5-expressing plasmids with HSV-1 infection” . Actually, this is a widely used and extensively recognized experimental method to verify the potential ISGs involved in virus-induced IFN pathway. There are many publications to use similar experimental design from both others group and our team [1-4]. For example, one study reported that KAT5 promoted innate immunity to DNA virus [1]. In their result “KAT5 Positively Regulates cGAS-Mediated Innate Immune Signaling”, they described “Overexpression of KAT5 promoted HSV-1–induced transcription of downstream genes. The stable KAT5-overexpressing and control THP-1 cell lines were left uninfected or infected with HSV-1 (multiplicity of infection [MOI] = 1) for 10 h before qPCR analysis”. Another study used consensus interferon (Con-IFN) to identify which ISGs involved in regulating IFN-βactivation [2]. In their Fig. 4: Screen for ISGs involved in SeV-induced IFN-β promoter activation, they said “HEK293T cells were transfected with the firefly luciferase-expressing reporter IFN-β-luc and the renilla luciferase-expressing control reporter TK-renilla, with or without a plasmid expressing the ISG indicated. At 24 h post-transfection, cells were infected with 10 HAU/mL SeV for 12 h.” Besides, a recent paper also used a similar approach to demonstrate the role of HDAC3 in innate immune positive regulation [3]. In their result “FOXK1 positively regulates type I IFN response”, they infected RAW264.7 with VSV to illustrate the FOXK1 positive regulation in the innate immunity pathway.
Thank you for your comments.
- To exclusively conclude the independency of the cGAS-STING pathway on induction of IFN-B by SerpinA5, it'd be better if authors add more control which is SerpinA5-expressing plasmids alone without any treatment. And, the standard protocol for assessing activation of the cGAS-STING pathway i.e. Chamma H, Guha S, Laguette N, Vila IK. Protocol to induce and assess cGAS-STING pathway activation in vitro. STAR Protoc. 2022 May 14;3(2):101384. doi: 10.1016/j.xpro.2022.101384. should be employed.
Answer:Thank you for your recommendation, and we have read this paper carefully. In this literature, cell was treated with dsDNA to activate the cGAS-STING pathway induced IFN-βproduction and the mRNA level of some molecules of the cGAS-STING pathway was evaluated by RT-qPCR. Consistent with this paper, in our study, to prove the independency of cGAS-STING signaling pathway in promotion of IFN-βproduction by SerpinA5, we conducted the coimmunoprecipitation assay to test the potential interaction between SerpinA5 and the key molecules of the cGAS-STING signaling pathway. Our results showed that serpina5 did not interact with cGAS, STING, TBK1, IRF3 and IRF7 (Fig. 3F). Moreover, we blocked the cGAS-STING signaling using C-176, which is a kind of STING inhibitor. Results showed that SerpinA5 overexpression can upregulate the expression of IFN-β and ISG56 even in presence of C-176 treatment (Fig.3G). Taken together, these findings support that SerpinA5 can activate the IFN production independent of cGAS-STING signaling pathway.
Minor comments:
1.Figure 1H, 1I, 1K and 1L, x-axis should be plasmid?
Answer: Thank you for your mention. The x-axis represents the transfection doses of SerpinA5 expressing plasmids, and we have described it in the figure legend in detail.
2.Line 200: please recheck "serpina5".
Answer: Thank you for your mention. We have rechecked it carefully.
Overall, we sincerely hope these data are convincing to support our conclusion. Thank you again for your kind comments and suggestions.
References:
- Song, Z.M., et al., KAT5 acetylates cGAS to promote innate immune response to DNA virus. Proc Natl Acad Sci U S A, 2020. 117(35): p. 21568-21575.
- Zhang, X., et al., Identification of new type I interferon-stimulated genes and investigation of their involvement in IFN-beta activation. Protein Cell, 2018. 9(9): p. 799-807.
- Yang, L., et al., Histone deacetylase 3 contributes to the antiviral innate immunity of macrophages by interacting with FOXK1 to regulate STAT1/2 transcription. Cell Rep, 2022. 38(4): p. 110302.
- Zhao,J., et al., Kynurenine-3-monooxygenase (KMO) broadly inhibits viral infections via triggering NMDAR/Ca2+ influx and CaMKII/ IRF3-mediated IFN-β production.PLoS Pathog. 2022. 18(3):e1010366.